# Serum Immunoglobulin G Is a Marker of Hidradenitis Suppurativa Disease Severity

**DOI:** 10.3390/ijms232213800

**Published:** 2022-11-09

**Authors:** Dillon Mintoff, Isabella Borg, Nikolai P. Pace

**Affiliations:** 1Department of Pathology, Faculty of Medicine and Surgery, University of Malta, Msida 2080, MSD, Malta; 2Department of Dermatology, Mater Dei Hospital, Msida 2080, MSD, Malta; 3Centre for Molecular Medicine and Biobanking, University of Malta, Msida 2080, MSD, Malta; 4Department of Pathology, Mater Dei Hospital, Msida 2080, MSD, Malta; 5Department of Anatomy, Faculty of Medicine and Surgery, University of Malta, Msida 2080, MSD, Malta

**Keywords:** hidradenitis suppurativa, inflammation, immunoglobulins, severity, biomarkers

## Abstract

Hidradenitis suppurativa (HS) is a chronic inflammatory condition of the skin that is brought about by autoinflammation and hyperkeratosis at the pilosebaceous unit. The clinical severity of HS can be measured using static (Hurley Severity Scoring (HSS)) and/or dynamic (International HS Severity Scoring System (IHS4)) severity scoring instruments. However, few clinically available serological parameters have been found to correlate with disease severity. In this study, we sought to investigate the role of serum immunoglobulin (Ig) G, M and A levels as biomarkers of disease severity and to compare them with other, more conventional inflammatory indices, such as the erythrocyte sedimentation rate, C-reactive protein, the neutrophil–lymphocyte ratio, the platelet–lymphocyte ratio and the systemic immune-inflammation index. In this cross-sectional study, patients were recruited from the only dermatology referral centre in Malta, Europe, and subjected to clinical examination and the assessment of inflammatory and immunologic parameters. Serum IgG, M and A levels were assessed using the Atellica^®^ NEPH 630 System (SIEMENS-Healthineers AF, Erlangen, Germany) nephelometric analyser. Serum IgG, M and A levels correlate with both dynamic and static HS severity scoring systems. Serum IgG behaves as a marker of severe HS disease as categorised by HSS and the IHS4. Our findings suggest that the serum IgG level can be used in the clinical setting as a biomarker of disease severity and, therefore, as an adjunct to clinical severity scoring.

## 1. Introduction

Hidradenitis suppurativa (HS) is a chronic, disabling skin disease with an estimated global prevalence of 0.4% [1]. The condition is the product of environmentally and genetically driven autoinflammation and hyperkeratinisation of the pilosebaceous unit (PSU) [2,3,4,5]. The primary lifestyle risk factors for HS are smoking and obesity. Additionally, pathogenic variation in genes encoding the ɣ-secretase complex, particularly *NCSTN*, are implicated in patients with familial forms of the disease [2,5,6,7]. HS is characterised by significant phenotypic diversity; however, all patients manifest painful nodules, abscesses and/or tunnels that periodically discharge their foul-smelling contents [8]. These lesions, predominantly found in intertriginous skin, occur in varying degrees of severity. HS has a profound impact on a patient’s quality of life and negatively affects their social, educational, sexual and vocational domains of living [9]. This burdensome disease is further compounded by a global diagnostic delay, limited FDA/EMA-approved therapeutic options and a paucity of clinically validated serological biomarkers [10,11].

Various severity scoring systems, broadly divided into static and dynamic scores, are available to determine clinical HS severity. The International Hidradenitis Suppurativa Severity Scoring System (IHS4) [12] and the Hurley Severity Scoring (HSS) [13] system are the most widely utilised dynamic and static scoring systems, respectively. Disparities in both inter- and intra-rater agreement and reliability hinder the reproducibility of clinical HS severity scoring, highlighting the need for serologically based, quantitative markers of disease severity [14,15].

Elevated serum immunoglobulin (Ig) levels have been established in other inflammatory conditions, including psoriasis, psoriatic arthropathy, atopic dermatitis and ankylosing spondylitis [16,17,18,19]. Given the correlation between nonspecific haematological markers of inflammation and HS disease severity [20,21,22,23], as well as the association of HS with an increased plasma cell population [24], we sought to assess whether serum IgG, M and/or A levels are surrogate serological indices of disease severity. To this end, we carried out a cross-sectional study in a population with a comparatively high prevalence of HS (Malta, Europe) [25].

## 2. Results

### 2.1. Population Characteristics

In total, 95 patients met the eligibility criteria for inclusion in this study. Table 1 presents the salient clinical and serological characteristics of the overall study cohort stratified according to disease severity using the Hurley and IHS4 classifications. No significant differences in gender or metabolic syndrome proportions between Hurley severity and IHS4 stage groups were observed. As expected, a significantly higher proportion of patients with severe disease were receiving treatment with the tumour necrosis factor (TNF)-α inhibitor adalimumab (χ^2^ test, *p* < 0.05).

### 2.2. Serum Inflammatory Markers Correlate with IHS4 Disease Severity

The relationship between serum IgG, M and A levels and haematological indices of inflammation was investigated by Spearman’s correlation. A correlation matrix is provided in Figure 1. Significant positive correlations were observed between serum IgG, IgM and IgA levels and disease severity as assessed by the IHS4 score. No differences in age, body mass index (BMI), leukocyte count, IgG, IgM and IgA levels, NLR, PLR, CRP or IHS4 were detected between genders (Mann–Whitney U test, *p* > 0.05).

### 2.3. Serum Immunoglobulin Levels Differ across HS Disease Severity

We evaluated the distribution of serum IgG, M and A levels against disease severity as categorised by IHS4 and Hurley classifications. A significant difference in IgG, IgM and IgA levels across disease severity by both Hurley and IHS4 classifications was observed (Kruskal–Wallis ANOVA, *p* < 0.05) (Figure 2), with higher levels in patients with severe disease when compared to those with mild disease. No differences in IgG, IgM or IgA levels between mild and moderate disease were observed.

Generalised linear models were applied to evaluate the association between IgG, M and A levels and disease severity using both Hurley and IHS4 classification criteria. In age- and gender-adjusted models (Model 1), serum IgG levels, but not IgA or IgM, were significantly associated with IHS4 (β = 0.135, 95% CI 0.056–0.213, *p* = 0.01). After further adjusting for multiple confounders (Model 2: age, gender, smoking and a diagnosis of metabolic syndrome; Model 3: Model 2 and additionally biological (adalimumab) therapy), a similar trend was observed between IHS4 scale and IgG levels, but not for IgM and IgA classes. Similarly, ordinal regression analysis with the Hurley stage as the categorical response variable identified an identical association between IgG levels and disease severity in all three models (Table 2).

### 2.4. Serum IgG Level Identifies Patients with Severe HS

Serum IgG levels demonstrated a strong discriminatory capacity to distinguish HS patients with severe disease according to the Hurley criteria (AUC = 0.90) from cases with mild or moderate disease severity and a moderate discriminatory capacity (AUC = 0.79) to distinguish cases with severe HS according to the IHS4 criteria from cases with mild or moderate disease (Figure 3).

The highest Youden index that best discriminates cases with severe (stage 3) Hurley disease from those with mild (stage 1) or moderate (stage 2) disease corresponds to an IgG of 13.4 g/L (sensitivity of 0.867; specificity of 0.823). This cut-off carries a high relative risk (RR = 8.0, 95% CI = 2.82 to 22.9, *p* = 0.0001) of severe disease compared to mild or moderate disease using the Hurley classification. On the other hand, the highest Youden index that best discriminates cases with severe disease according to the IHS4 criteria from those with mild or moderate disease corresponds to an IgG of 12.1 g/L (sensitivity of 0.895; specificity of 0.587). This cut-off carries a high relative risk (RR = 5.3, 95% CI = 1.66 to 17.7, *p* = 0.005) of severe disease compared to mild or moderate disease using the IHS4 classification.

When compared to IgM, IgA, NLR and SII, serum IgG was shown to be the superior identifier of patients with Hurley and IHS4 severe disease (Figure 4, Table 3).

We also explored the sensitivity of the analysis to the exclusion of subjects having moderate disease severity. IgG levels demonstrated a strong discriminatory capacity to distinguish participants with severe from mild disease according to the Hurley staging system (AUC = 0.92, CI 0.856–0.992, *p* < 0.01) and according to the IHS4 category (AUC = 0.81, CI 0.697–0.917, *p* < 0.01) (Table 3). When contrasting patients with moderate vs. mild disease according to both Hurley and IHS4 classifications, IgG, M and A levels and inflammatory indices did not exceed significance thresholds.

## 3. Discussion

This study evaluates the association between IgG, IgM and IgA serum levels and HS disease severity. We show that the serum levels of these immunoglobulin subtypes correlate with HS severity and support the role of the serum IgG level as a disease severity biomarker with superior performance compared to other haematological inflammatory indices of inflammation, such as the SII and NLR. To the best of our knowledge, this is the first study to explore the association between serum immunoglobulin G, M and A levels and HS severity.

The role of plasma cells and B cells in the pathophysiology of HS has been explored, with both cell types predominating in HS skin lesions [24,26]. Furthermore, HS lesional skin biopsies were shown to be rich in IgG immune complex deposits and to upregulate several immunoglobulin genes (including *IGLV3-26* (immunoglobulin lambda variable 3–26)*, CD19* and *CD79a*) [23,24]. IgG immune complex and complement deposition has been shown not only to be prominent in HS lesional skin but also to sustain inflammation [24]. Of the leukocyte population, plasma cells show the most pronounced difference in concentration across individual HS severity groups [27], and HS disease progression is characterised by a shifting of skin-infiltrating memory B cells to plasma cells [26]. The presence of plasma cells in the dermis in more severe forms of HS may represent immune activation towards the infiltration of the dermis by keratinocytes. As part of the “epimmunome”, products of infiltrating keratinocytes may exert an effect on both the innate and adaptive immune cell populations [28,29]. HS keratinocytes are known to exhibit an abnormal cytokine response upon being triggered [30].

The total lesional leukocyte count has been shown to differ between HS disease categories as defined by histological severity: “early” when the epithelium is intact (distended but unruptured follicular epithelium) and “late” (with migrating stratified squamous epithelium from ruptured follicles and epithelial sheets) when it is not [27]. These results have been replicated, with plasma cells scattered in advanced HS lesions typified by intradermal migrating epithelial sheets [31]. Skin exhibiting migrating stratified squamous epithelium from ruptured follicles has the greatest number of plasma cells, as opposed to healthy control skin, which has the lowest concentration of this leukocyte subtype [27]. The HS-associated dysregulation of NOTCH signalling may potentiate dermal invasion by keratinocytes, leading to an immune response [32,33]. Taken together, these pathophysiological processes may account for the fact that serum IgG levels only confidently discriminate severe HS (as defined by HSS and IHS4) from milder forms of the disease.

The role of serum IgG in the pathophysiology of HS may also extend to the activation of the complement cascade (through the classical pathway), which has been gaining significance in HS research and as a druggable pathway [34,35]. Single-cell RNA sequencing studies have shown the upregulation of genes in monocytes and macrophages that suggest polarisation to an M1 macrophage phenotype [36], which mediates antibody-dependent cellular toxicity leading to marked inflammation [35]. This suggests a potential link between plasma cells, Igs and complement and the heavy inflammatory burden in HS.

It is interesting to note that serum IgA levels were also elevated in HS patients with severe disease but did not exceed significance thresholds in regression models. There is a paucity of data with regard to the biological relevance of serum (and secreted) IgA in HS patients. Cardiometabolic risk factors and metabolic syndrome have been positively associated with serum IgA levels in the general adult population [37]. Both HS and metabolic syndrome are characterised by a shared overlap in patient phenotypes and a subclinical chronic proinflammatory state, driven by adipocytokine signalling and M1 macrophage activation [38]. Plausibly, metabolic syndrome and its constituent components represent latent confounders in our analysis, and longitudinal cohort studies are required to evaluate cause–effect relationships in this context. More recently, serum anti-Carboxy-ethyl-lysine (CEL) IgA and IgG antibody titres were found to be elevated in an HS cohort. Anti-CEL IgG levels also correlated with HS disease severity [39]. Furthermore, serum titres of anti-*Saccharomyces cerevisiae* (ASCA) IgA and IgG were found to be elevated in HS patients and are postulated to be a biomarker of disease severity [40] as well as phenotype [41]. With regard to nonspecific markers of inflammation, proportional increases in NLR and SII across Hurley severity scores have already been identified [42]. Studies with a paucity of patients suffering from Hurley stage 1 disease failed to establish this correlation [43].

The findings from this study warrant interpretation in the context of some limitations. Primarily, no data on key inflammatory markers, such as high-sensitivity CRP and cytokine levels, were available. IgG subclasses have not been investigated in this study. The cross-sectional design limits the evaluation of the interaction between immunoglobulin levels and clinical outcomes. It is acknowledged that a single measurement of immunoglobulin levels does not fully capture the systemic proinflammatory status in HS, particularly with regard to disease dynamics and the response to pharmacotherapy. No causal direction between Ig levels and HS severity can be robustly inferred, and our findings provide no direct insight into causative mechanisms. Furthermore, replication in different cohorts to assess the clinical utility and validity of Ig levels as a disease severity biomarker is required.

## 4. Materials and Methods

### 4.1. Study Cohort

We conducted a single-centre, cross-sectional study involving adults of Maltese-Caucasian ethnicity. Participants were enrolled from the dermatology outpatient department of the referral centre by a dermatologist experienced in the diagnosis and management of HS. This centre is the exclusive referral point for HS specialist care in the Maltese archipelago, catering to a population of approximately 550,000 inhabitants. A diagnosis of HS was made (or reaffirmed) in adults based on the Dessau Definition of HS, namely: “recurrent, painful or purulent, deep-seated lesions occurring more than twice every six months, located in the groin, axilla, perineum, buttocks or sub mammary folds” [8]. The following exclusion criteria were applied:Patients diagnosed with or suspected to have an active solid organ or haematological malignancy or terminal illness;Pregnancy;Patients found to have a monoclonal gammopathy;Patients diagnosed or suspected to have other active, systemic infective or inflammatory processes;Patients diagnosed or suspected to have a primary immunodeficiency disorder;Patients with suspected or confirmed seropositive coagulopathy;Patients treated with a systemic antibiotic, non-biologic immunomodulatory/immunosuppressant therapy and/or intralesional and/or oral corticosteroid in the last six weeks;Patients who underwent surgical intervention in the last three months;Patients who were unable to provide their own voluntary informed consent;Patients who could not attend timely follow-up appointments.

Prospective participants fitting the above inclusion and exclusion criteria were invited to participate in the study between January and June 2022.

Detailed medical history taking and clinical examination were carried out. Anthropometric measurements were recorded with the participants dressed in light clothing and without shoes using the same validated equipment, which was calibrated in accordance with WHO recommendations. Body weight was measured to the nearest 0.1 kg, and height and waist circumference were measured to the nearest 0.1 cm.

Patients were diagnosed with metabolic syndrome according to the criteria defined by the National Cholesterol Education Program expert panel on the detection, evaluation and treatment of high blood cholesterol in adults (NCEP-ATP III) [44].

### 4.2. HS Severity Staging

HS disease severity was classified according to both HSS and IHS4. Patients were categorised into one of the three Hurley severity stages as defined by Hurley (1989) [13]:Stage I (mild): single or multiple nodules or abscesses in a typical site without the presence of sinus tracts;Stage II (moderate): single or multiple, recurrent, widely separated nodules and/or abscesses and tunnels with scarring;Stage III (severe): diffuse nodules, abscesses and draining tunnels involving the majority of an affected area.

IHS4 scores were calculated as defined by Zouboulis et al. [12] as the sum of the number of nodules multiplied by 1, the number of abscesses multiplied by 2 and the number of draining tunnels multiplied by 4. HS severity was categorised according to the following IHS4 cut-off scores:Mild HS: for scores ≤ 3;Moderate HS: for scores ranging from 4 to 10;Severe HS: for scores ≥ 11.

### 4.3. Assessment of Serum Immunoglobulin Levels

Patients were fasted for a minimum of 12 h before bloodletting. Blood was drawn from prospective participants for serological assessment of biochemical, immunological and haematological parameters. Blood samples were analysed within an hour of collection.

Serum Ig levels and Ig classes were quantified using the Atellica^®^ NEPH 630 System (SIEMENS-Healthineers AF, Erlangen, Germany) nephelometric analyser according to the manufacturer’s instructions. The following reference intervals for Ig classes were applied for serum samples: IgG 7.0–16 g/L, IgM 0.4–2.3 g/L and IgA 0.7–4.0 g/L [45]. Samples with IgG levels of >16.0 g/L and/or IgM levels >2.3 g/L serum and/or IgA levels of >4.0 g/L were referred for serum protein electrophoresis (SPE) to assess for a monoclonal band and to elucidate Ig subclasses. Patients with an elevated ɣ globulin fraction but in whom a monoclonal band was not detected on SPE were assessed for evidence of κ and λ free light chain (FLC) monoclonality. The reference ranges for κ FLC and λ FLC were 6.7–22.4 mg/L and 8.3–27.0 mg/L, respectively [46]. Patients with elevated IgA levels were investigated for IgA anti-endomysial antibodies. Patients with elevated Ig levels and a personal or family history of thrombosis were assessed for serum IgM and IgA anti-cardiolipin antibodies.

### 4.4. Calculation of Inflammatory Parameters

The following surrogate indices of systemic inflammation were derived from routine complete blood counts, inclusive of the leukocyte differential count. These derived indices have been associated with HS severity in other publications. The erythrocyte sedimentation rate (ESR) and C-reactive protein (CRP) were quantified using standard hospital auto-analysers.

Neutrophil–lymphocyte ratio (NLR) (neutrophil count divided by lymphocyte count);Platelet–lymphocyte ratio (PLR) (platelet count divided by lymphocyte count);Monocyte-lymphocyte ratio (MLR) (monocyte count divided by lymphocyte count);Lymphocyte–monocyte ratio (LMR) (lymphocyte count divided by monocyte count);Systemic immune-inflammation index (SII) (platelet x neutrophil/lymphocyte count [47]).

### 4.5. Statistical Analysis

The normality of continuous variables was assessed by the Shapiro–Wilk and Kolmogorov–Smirnoff tests. All continuous parameters exhibited a skewed non-normal distribution, and non-parametric statistics with medians and interquartile ranges are presented. Categorical variables are presented as percentages, and χ^2^/Fisher’s exact tests were applied to compare dichotomous outcomes. Spearman’s rank-order coefficient was used to explore the strength and direction of association between quantitative variables. To evaluate differences in quantitative variables between groups, the Kruskal–Wallis test was used for comparison between three or more categories, followed by Dunn’s post hoc test for pairwise comparison between subgroups. The independent-samples Mann–Whitney U test was used for comparison between two categories.

The effect of Ig levels on disease severity as assessed by the IHS4 scale was evaluated using generalised linear modelling, with adjustment for multiple confounders. Ig levels were the independent predictors, and IHS4 was the continuous dependent response variable. Generalised linear modelling specifying gamma as the distribution and Log as the link function was applied in view of the positively skewed distributions of the parameters under investigation. Multicollinearity diagnostics revealed no dependency between the independent variables, with variance inflation factors <2, thus indicating they could be reliably used as independent predictors in the model. The relationship between Ig levels and disease severity according to HSS was evaluated by ordinal regression analysis with the Hurley stage as the categorical response variable.

Adjustment for confounding factors was performed as follows. In Model 1, adjustment for age and gender was applied. In Model 2, we adjusted for age, gender, smoking and a diagnosis of metabolic syndrome. In Model 3, we further adjusted for biological therapy (namely, TNFα inhibitor: adalimumab) in addition to the parameters incorporated in Model 2.

Receiver operating characteristic (ROC) analysis was used to compute the area under the curve (AUC) to assess the performance of selected inflammatory indices and serum IgG, IgM and IgA levels in discriminating cases with severe disease (IHS4 > 11 and Hurley stage 3). In the primary analysis, subjects with mild or moderate disease according to the Hurley or IHS4 classifications were considered an aggregate category. The highest Youden index (sensitivity + specificity − 1) was used to determine optimal cut-off points. In the secondary analysis, subjects with moderate disease were excluded, and ROC analysis was applied to directly contrast severe vs. mild disease. ROC analysis was performed using the easyROC R application, and cut-off values were determined using the OptimalCutpoints R package [48,49]. All analyses were performed using IBM SPSS version 26 and R v.3.4.2.

## 5. Conclusions

In conclusion, this study complements previous transcriptomic and immunohistochemical studies that have identified elevated immunoglobulin levels in HS patients. These data provide the groundwork for a quantifiable, putative link between the observed deposition of Ig complexes in lesional HS skin and a shift towards a plasma-cell-heavy population with worsening HS disease severity. Further studies aimed at evaluating the lymphocyte immunophenotype of HS in both cutaneous tissue and peripheral blood and relating the findings to pathological mechanisms via multi-omic approaches should be considered.

## Figures and Tables

**Figure 1 ijms-23-13800-f001:**
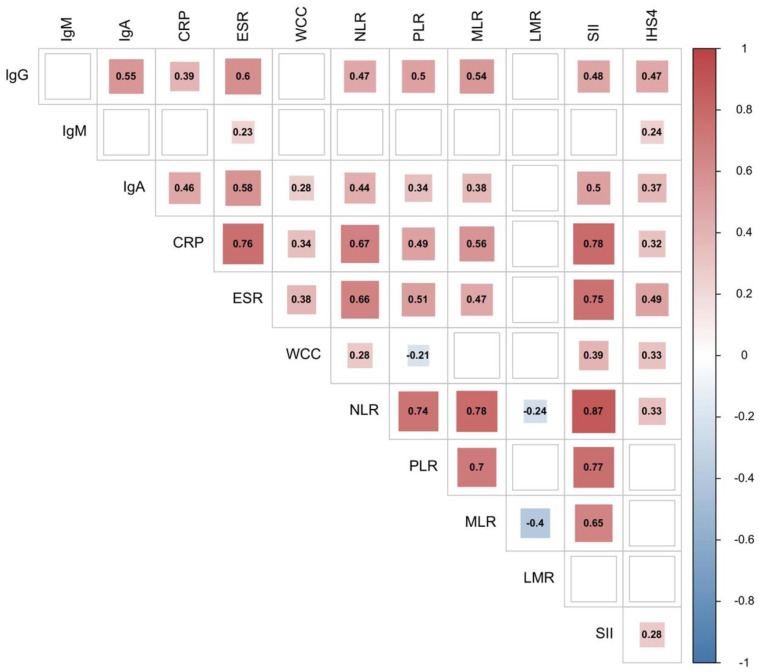
Correlation matrix between immunoglobulin levels, haematological indices of inflammation and disease severity according to IHS4 classification. The square colour depicts Spearman’s rank-order correlation coefficient, with its size and colour intensity indicating the magnitude of the correlation coefficient. Significant correlation coefficients are labelled. Empty cells represent insignificant pairwise correlation between indices. SII—systemic immune-inflammation index.

**Figure 2 ijms-23-13800-f002:**
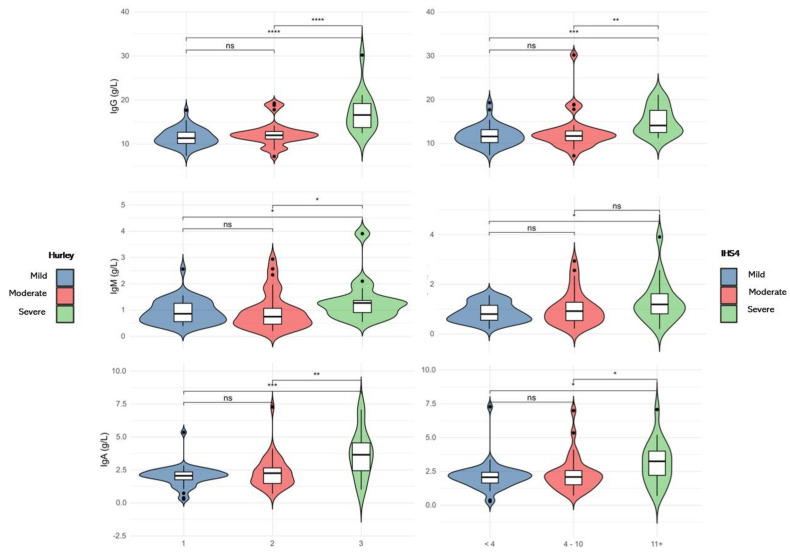
Violin plots illustrating immunoglobulin levels across HS severities as assessed by the Hurley severity score (left panels) and International Hidradenitis Suppurativa Severity Scoring System (IHS4) (right panels) severity staging. The violin plots depict data distribution. The centre line in the box-plot illustrates the median, the box limits indicate the 25th and 75th percentiles and the whiskers extend to 1.5 times the interquartile range from the 25th and 75th percentiles. * *p* < 0.05, *** p* < 0.01, **** p* < 0.001, ***** p* < 0.0001, ns—not significant.

**Figure 3 ijms-23-13800-f003:**
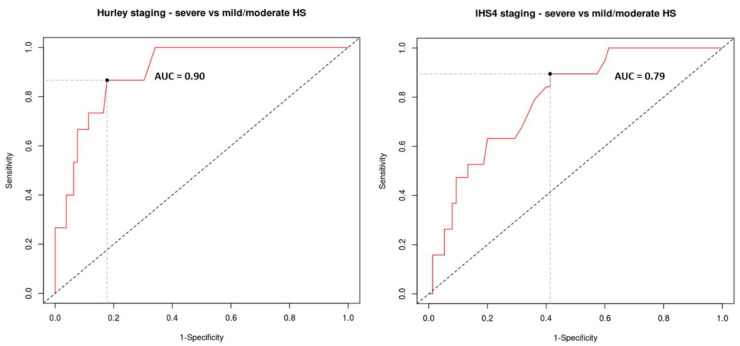
Serum IgG levels demonstrate a strong discriminatory capacity to identify HS patients with Hurley stage 3 disease from stage 1 and 2 disease (left, AUC 0.9) and a moderate discriminatory capacity (right, AUC 0.79) to identify HS patients with IHS4 “severe” disease from patients with mild and moderate disease as classified by IHS4. Youden’s index is indicated by the filled circle. AUC = area under the curve.

**Figure 4 ijms-23-13800-f004:**
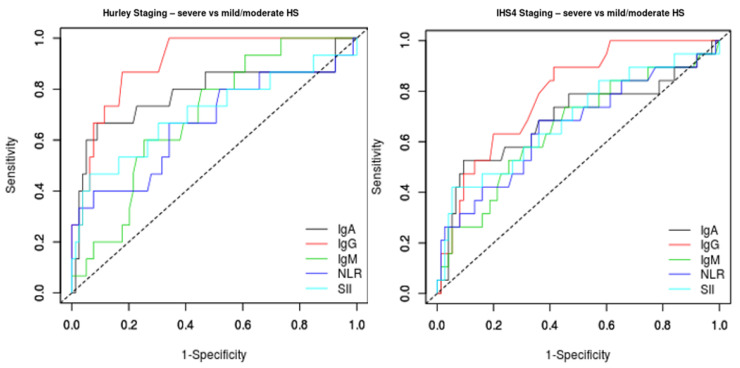
Comparison of ROC curves for IgA, IgG, IgM, neutrophil–lymphocyte ratio (NLR) and systemic immune-inflammation index (SII), with severe disease as the status variable. IgG levels demonstrate stronger discriminatory value for severe disease when compared to other blood-based indices of inflammation, such as the NLR and SII.

**Table 1 ijms-23-13800-t001:** The salient clinical and serological characteristics of the study cohort. Data substratified by disease severity as defined by the Hurley and IHS4 classifications are also presented. The median and interquartile ranges (in brackets) are presented for continuous variables. *Rx = treatment; MetS = metabolic syndrome*; *BMI = body mass index; WBC = white blood count; NLR = neutrophil–lymphocyte ratio; PLR = platelet–lymphocyte ratio; MLR = monocyte–lymphocyte ratio; LMR = lymphocyte–monocyte ratio; ESR = erythrocyte sedimentation rate; CRP = C-reactive protein; IHS4 = International Hidradenitis Suppurativa Severity Scoring System*.

		Hurley Stage				IHS4 Stage			
	Overall *(n = 95)*	Mild *(n = 45)*	Moderate *(n = 35)*	Severe *(n = 15)*	*p* Value	Mild *(n = 43)*	Moderate *(n = 33)*	Severe *(n = 19)*	*p* Value
**% Females**	50.5	48.9	48.6	60	0.759	53.5	42.4	57.9	0.489
**% Current and Ex-smokers**	63.1	51.1	77.1	46.7	**0.029**	55.9	72.7	43.2	0.209
**% Rx adalimumab**	14.7	6.7	11.4	46.7	**0.003**	9.3	12.1	31.6	**0.04**
**% MetS**	34	34.1	34.3	33.3	1	30.2	46.9	21.1	0.14
**Age *years***	34 (24)	29 (16)	37 (21)	45 (31)	0.087	30 (22)	35 (14)	36 (31)	0.433
**BMI kg/m^2^**	31.3 (11)	33.6 (12.3)	29.4 (11.4)	33.9 (12.6)	0.188	28.9 (11.9)	34.1 (12.6)	31.5 (9)	0.207
**WBC × 10^9^/L**	7.97 (2.86)	7.34 (3.07)	7.98 (2.87)	9.14 (2.67)	**0.014**	7.3 (3.54)	7.92 (2.21)	8.78 (2.89)	**0.05**
**NLR**	2.32 (1.04)	2.11 (1.05)	2.36 (0.94)	2.66 (3.01)	**0.05**	2.15 (1.07)	2.38 (0.97)	2.66 (2.05)	**0.046**
**PLR**	136.42 (73.59)	135.40 (67.34)	140.10 (59.50)	167.89 (133.08)	0.768	136.52 (72.41)	138.57 (73.59)	132.17 (97.56)	0.919
**MLR**	0.27 (0.12)	0.27 (0.10)	0.29 (0.12)	0.28 (0.27)	0.746	0.26 (0.1)	0.29 (0.13)	0.28 (0.13)	0.445
**LMR**	3.66 (1.69)	3.73 (1.53)	3.43 (1.70)	3.62 (3.13)	0.746	3.87 (1.65)	3.44 (1.87)	3.54 (1.8)	0.445
**ESR mm 1st *Hr***	14 (26)	9 (15)	17 (28)	46 (45)	**<0.001**	9 (12)	14 (21)	36 (38)	**<0.001**
**CRP mg/L**	3.9 (7)	2.1 (3.2)	5 (7.6)	12 (21.5)	**<0.001**	2 (4.2)	4.7 (6.6)	9.8 (21.3)	**0.001**
**Ferritin ng/mL**	61 (90)	62 (81)	60 (98)	56 (143)	0.847	55 (76)	52 (130)	70 (60)	0.625
**IgG g/L**	12.2 (2.8)	11.35 (2.7)	12 (1.8)	16.6 (6.7)	**<0.001**	11.6 (3.1)	11.7 (2.4)	14.1 (5.5)	**<0.001**
**IgM g/L**	0.9 (0.73)	0.86 (0.7)	0.75 (0.61)	1.27 (0.51)	**0.05**	0.8 (0.62)	0.92 (0.77)	1.19 (1.11)	0.1
**IgA g/L**	2.195 (1)	2.055 (0.65)	2.253 (1.25)	3.65 (2.39)	**0.002**	2.07 (0.83)	2.09 (1.08)	3.25 (1.97)	**0.036**
**IHS4**	4 (8)	2 (3)	6 (8)	24 (15)	**<0.001**	1 (2)	5 (3)	18 (15)	**<0.001**
**% Hurley Stage**		47.4	36.8	15.8					

**Table 2 ijms-23-13800-t002:** Generalised linear models evaluating relationship between IgG, A and M level subtypes and International Hidradenitis Suppurativa Severity Scoring System (IHS4) and Hurley Severity Scoring (HSS) after adjusting for Models 1, 2 and 3.

		IHS4		HSS	
		β (95% CI)	*p* Value	β (95% CI)	*p* Value
**Model 1**	**IgG**	0.14 (0.041–0.22)	**0.001**	0.31 (0.146–0.475)	**<0.001**
	IgA	0.109 (−0.09–0.307)	0.284	0.354 (−0.042–0.749)	0.08
	IgM	0.237 (−0.049–0.542)	0.104	0.303 (−0.089–0.7649)	0.13
**Model 2**	**IgG**	0.14 (0.058–0.223)	**0.001**	0.3847 (0.192–0.581)	**<0.001**
	IgA	0.081 (−0.124–0.285)	0.438	0.291 (−0.107–0.689)	0.151
	IgM	0.281 (−0.02–0.583)	0.06	0.422 (−0.004–0.847)	0.052
**Model 3**	**IgG**	0.125 (0.038–0.213)	**0.005**	0.373 (0.173–0.574)	**<0.001**
	IgA	0.034 (−0.186–0.255)	0.76	0.248 (0.170–0.667)	0.245
	IgM	0.274 (−0.022–0.571)	0.07	0.398 (−0.037–0.833)	0.073

**Table 3 ijms-23-13800-t003:** Comparison of area under the curve (AUC) values for IgG IgM, IgA, NLR and SII, with severe disease as the status variable. IgG levels demonstrated stronger discriminatory value for severe disease when compared to other blood-based indices of inflammation, such as the NLR and SII, using both Hurley and IHS4 classifications. The top panel depicts receiver operating characteristic (ROC) analysis with subjects having mild and moderate disease, considered an aggregate reference category. Exclusion of subjects with moderate disease (bottom panel) yielded consistent findings.

	Hurley Severe vs. Hurley Moderate/Mild					IHS4 Severe vs. IHS4 Moderate/Mild				
Marker	AUC	SE AUC	95%CI L	95%CI U	*p*	AUC	SE AUC	95%CI L	95%CI U	*p*
**IgG**	0.90	0.04	0.86	0.97	<0.001	0.79	0.05	0.69	0.89	<0.001
**IgM**	0.69	0.07	0.56	0.82	<0.001	0.65	0.08	0.51	0.80	0.409
**IgA**	0.78	0.08	0.62	0.95	<0.001	0.69	0.08	0.54	0.84	0.011
**SII**	0.71	0.09	0.53	0.88	0.020	0.69	0.08	0.53	0.85	0.017
**NLR**	0.67	0.09	0.49	0.84	0.058	0.66	0.08	0.51	0.81	0.041
	**Hurley Severe vs. Hurley Mild**					**IHS4 Severe vs. IHS4 Mild**				
**IgG**	0.92	0.04	0.86	0.99	<0.001	0.81	0.06	0.70	0.92	<0.001
**IgM**	0.69	0.08	0.54	0.84	0.030	0.68	0.08	0.52	0.83	0.026
**IgA**	0.80	0.08	0.64	0.97	<0.001	0.71	0.08	0.54	0.87	0.010
**SII**	0.74	0.09	0.56	0.94	0.007	0.69	0.08	0.54	0.84	0.018
**NLR**	0.70	0.09	0.53	0.87	0.024	0.69	0.08	0.54	0.84	0.019

## Data Availability

Data are available from D.M. upon reasonable request.

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
