# Peer review of "Serum Immunoglobulin G Is a Marker of Hidradenitis Suppurativa Disease Severity"

_ijms, 2022, doi:10.3390/ijms232213800_

Round 1
Reviewer 1 Report
In the article 'Serum Immunoglobulin G is a marker of Hidradenitis Suppurativa disease severity' authors study immunoglobulin levels in patients with HS. Methods and exclusion criteria are solid and the article is interesting. Recently B cell and plasma cell have been shown increased in lesional skin of HS patients. This finding may indicate presence of autoantibodies in HS patients. The authors show the elevated levels of IgG and in some aspect IgA higher in HS patients. I have some comments.
Minor:
1-Authors can include elevated IgA levels in the discussion.
2-Plasma cells in the lesion indicates production of antibodies against an antigen or several antigens. Authors could potentially shortly discuss the antiboides detected in this group of patienst. Such as: Anti saccharomyces ab (IgG and IgA) and anti-CEL antibodies are found in HS patients.
https://pubmed.ncbi.nlm.nih.gov/36116506/
https://pubmed.ncbi.nlm.nih.gov/32482529/
3-I wonder if any particular IgG group was elevated. (IgG1-2-3-4?). Although this is not investigated in this study, it can be further tackled in a new study and lack of this information can be added to limitations.
Author Response
Dear reviewer, thank you for constructive comments and your overall positive feedback of the manuscript.
Minor:
1-Authors can include elevated IgA levels in the discussion.
2-Plasma cells in the lesion indicates production of antibodies against an antigen or several antigens. Authors could potentially shortly discuss the antiboides detected in this group of patienst. Such as: Anti saccharomyces ab (IgG and IgA) and anti-CEL antibodies are found in HS patients.
https://pubmed.ncbi.nlm.nih.gov/36116506/
https://pubmed.ncbi.nlm.nih.gov/32482529/
Reply: Thank you for these vauluable suggestions. We have tackled both recommendations in the discussion;
It is interesting to note that serum IgA levels were also elevated in HS patients with severe disease, but did not exceed significance thresholds in regression models. There is a paucity of data with regards to the biological relevance of serum (and secreted) IgA in HS patients. Cardiometabolic risk factors and the metabolic syndrome have been positively associated with serum IgA levels in the general adult population [37]. HS and the metabolic syndrome are characterised a shared overlap in patient phenotypes and a subclinical chronic proinflammatory state, driven by adipocytokine signalling and M1 macrophage activation. [38] Plausibly, metabolic syndrome and its constituent components represent latent confounders in our analysis, and longitudinal cohort studies are required to evaluate cause-effect relationships in this context. More recently, serum anti-Carboxy-ethyl-lysine (CEL) IgA, and IgG, antibody titres were found to be elevated in an HS cohort. Anti-CEL IgG levels also correlated with HS disease severity[39]. Furthermore, serum titres of anti-Saccharomyces cerevisiae (ASCA) IgA and IgG were found to be elevated in HS patients, and postulated to be a biomarker of disease severity [40] as well as phenotype[41].
3-I wonder if any particular IgG group was elevated. (IgG1-2-3-4?). Although this is not investigated in this study, it can be further tackled in a new study and lack of this information can be added to limitations.
Reply: Thank you for this interesting observation. We agree that this will have to be tackled in another study and have thus mentioned in the limitations.
Reviewer 2 Report
The manuscript „Serum immunoglobulin G is a marker of hidradenitis suppurativa disease severity“ presents original work and valuable results obtained by this group of authors. Generally, I consider this manuscript useful for current knowledge and data on this subject.
However, there are few suggestions:
-Please emphasize the method used in this study for the determination of immunoglobulins, including their mention in the Abstract.
- You commonly mention the general term “immunoglobulins”, but be specific (on each place), since there are different types of immunoglobulins (IgG, IgM, IgA, IgD, and IgE). So, in many places you think only on IgG.
- Also, below the tables, please add the abbreviations for better and more clear insight into the examined factors
-Mentoning the limitations and strengths of the study is more suitable for the end of Discussion instead in the Conclusion
-Authors could add some additional information/data form previous study as a potential explanation for the association between IgG and skin lesions – to support why serum IgG is a marker of Hidradenitis Suppurativa disease severity.
-References are written with different types of initial letters i.e. there are mixed capitals and small letters (uppercase and lowercase letters).
-Since I am a dermatologist, I can comment the text as a clinician, and the suggestions by the expert for statistics would be welcome.
(FEW MINOR REVISIONS ARE NEEDED)
Author Response
The manuscript „Serum immunoglobulin G is a marker of hidradenitis suppurativa disease severity“ presents original work and valuable results obtained by this group of authors. Generally, I consider this manuscript useful for current knowledge and data on this subject.
However, there are few suggestions:
Reply: Thank you for your thorough review of our manuscript We appreciate the relevant suggestions that you have made and which we have tackled.
-Please emphasize the method used in this study for the determination of immunoglobulins, including their mention in the Abstract.
Reply: Thank you, we have included the following sentence within the “methods” sub-section of the abstract.
Serum Ig G, M and A levels were assessed using the Atellica ® NEPH 630 System (SIEMENS-Healthineers AF, Erlangen, Germany) nephelometric analyser
We have highlighted this within the “methods” section of the manuscript as well.
- You commonly mention the general term “immunoglobulins”, but be specific (on each place), since there are different types of immunoglobulins (IgG, IgM, IgA, IgD, and IgE). So, in many places you think only on IgG.
Reply: thank you for picking this up. We have made the necessary adjustments throughout the text.
- Also, below the tables, please add the abbreviations for better and more clear insight into the examined factors
Reply: thank you, we have defined the abbreviations used in tables and figures within the table and figure legends at the first instance.
-Mentoning the limitations and strengths of the study is more suitable for the end of Discussion instead in the Conclusion
Reply: Thank you for this recommendation. We agree, and have therefore moved the limitations from the conclusion to the discussion.
-Authors could add some additional information/data form previous study as a potential explanation for the association between IgG and skin lesions – to support why serum IgG is a marker of Hidradenitis Suppurativa disease severity.
Reply: Thank you for this suggestion. We have expanded the discussion to include two previous studies (one of which was published in the last month) which correlate specific immunoglobulin antibodies to disease severity. There is a general paucity of data with regards to Immunoglobulin levels and HS disease severity, and that was one of the main drivers of this research endeavour.
It is interesting to note that serum IgA levels were also elevated in HS patients with severe disease, but did not exceed significance thresholds in regression models. There is a paucity of data with regards to the biological relevance of serum (and secreted) IgA in HS patients. Cardiometabolic risk factors and the metabolic syndrome have been positively associated with serum IgA levels in the general adult population [37]. HS and the metabolic syndrome are characterised a shared overlap in patient phenotypes and a subclinical chronic proinflammatory state, driven by adipocytokine signalling and M1 macrophage activation. [38] Plausibly, metabolic syndrome and its constituent components represent latent confounders in our analysis, and longitudinal cohort studies are required to evaluate cause-effect relationships in this context. More recently, serum anti-Carboxy-ethyl-lysine (CEL) IgA, and IgG, antibody titres were found to be elevated in an HS cohort. Anti-CEL IgG levels also correlated with HS disease severity[39]. Furthermore, serum titres of anti-Saccharomyces cerevisiae (ASCA) IgA and IgG were found to be elevated in HS patients, and postulated to be a biomarker of disease severity [40] as well as phenotype[41].
-References are written with different types of initial letters i.e. there are mixed capitals and small letters (uppercase and lowercase letters).
Reply: thank you for picking this syntax issue. We have used a professional citation software which detects the manuscript titles as defined by the publishing journal. Should the manuscript be accepted, we will work with the editorial production office to limit any incongruencies as requested.